# Detection of Chili Foreign Objects Using Hyperspectral Imaging Combined with Chemometric and Target Detection Algorithms

**DOI:** 10.3390/foods12132618

**Published:** 2023-07-06

**Authors:** Zhan Shu, Xiong Li, Yande Liu

**Affiliations:** School of Mechatronics & Vehicle Engineering, East China Jiaotong University, Nanchang 330013, China; suzhoumingyu@foxmail.com (Z.S.); LX15797652860@163.com (X.L.)

**Keywords:** foreign objects, hyperspectral imaging, object detection, spectral classification

## Abstract

Chilies undergo multiple stages from field production to reaching consumers, making them susceptible to contamination with foreign materials. Visually similar foreign materials are difficult to detect manually or using color sorting machines, which increases the risk of their presence in the market, potentially affecting consumer health. This paper aims to enhance the detection of visually similar foreign materials in chilies using hyperspectral technology, employing object detection algorithms for fast and accurate identification and localization to ensure food safety. First, the samples were scanned using a hyperspectral camera to obtain hyperspectral image information. Next, a spectral pattern recognition algorithm was used to classify the pixels in the images. Pixels belonging to the same class were assigned the same color, enhancing the visibility of foreign object targets. Finally, an object detection algorithm was employed to recognize the enhanced images and identify the presence of foreign objects. Random forest (RF), support vector machine (SVM), and minimum distance classification algorithms were used to enhance the hyperspectral images of the samples. Among them, RF algorithm showed the best performance, achieving an overall recognition accuracy of up to 86% for randomly selected pixel samples. Subsequently, the enhanced targets were identified using object detection algorithms including R-CNN, Faster R-CNN, and YoloV5. YoloV5 exhibited a recognition rate of over 96% for foreign objects, with the shortest detection time of approximately 12 ms. This study demonstrates that the combination of hyperspectral imaging technology, spectral pattern recognition techniques, and object detection algorithms can accurately and rapidly detect challenging foreign objects in chili peppers, including red stones, red plastics, red fabrics, and red paper. It provides a theoretical reference for online batch detection of chili pepper products, which is of significant importance for enhancing the overall quality of chili pepper products. Furthermore, the detection of foreign objects in similar particulate food items also holds reference value.

## 1. Introduction

Chili peppers belong to the Solanaceae family and are native to Mexico [1]. They were initially cultivated and used by indigenous peoples in Central and South America [2]. Since the 16th century, chili peppers have been brought to Europe and Asia, quickly becoming an essential seasoning in local cuisines [3]. Chili peppers are rich in nutrients such as vitamin C, vitamin A, fiber, and antioxidants [4]. Additionally, chili peppers have many health benefits, including promoting metabolism, enhancing the immune system, and reducing inflammation [5,6].

Chili pepper foreign bodies encompass various types, such as stones, fabrics, plastics, and more. Consuming chili peppers with foreign bodies can have adverse effects on human health. If these foreign bodies are not promptly detected and removed, they can have negative impacts on the oral cavity, esophagus, gastrointestinal tract, and other areas of the body [7]. Ingesting foreign bodies can cause abrasions to the oral and esophageal mucosa, leading to symptoms such as oral ulcers and bleeding. In severe cases, it can even result in esophageal perforation or bleeding [8]. Additionally, foreign bodies can carry pathogens or harmful substances, potentially causing poisoning or infection after entering the body [9].

Therefore, in the food production process, it is desirable to identify and locate foreign bodies through a non-destructive detection system before processing. Based on the information regarding the type and coordinates of the detected foreign bodies, appropriate measures can be taken by the operators to remove them, ensuring the safety of the food product.

In recent years, spectral imaging technology has been increasingly applied in the field of food inspection due to its advantages of speed, non-destructiveness, and accuracy [10]. Díaz et al. [11] evaluated the feasibility of this technology in detecting foreign bodies in meat products using a hyperspectral system, demonstrating its significant potential since it combines the advantages of traditional machine vision and spectroscopy. Saeidan et al. [12] investigated the feasibility of using hyperspectral imaging technology to detect and differentiate four types of foreign bodies (wood, plastic, stone, and plant organs) in the cocoa processing industry, and the results showed that the accuracy of the support vector machine (SVM) classifier in classifying cocoa beans and foreign bodies reached 89.10%. Sun et al. [13] developed an electromagnetic vibration feeder combined with a hyperspectral imaging system for detecting tea stems and insect foreign bodies in finished tea products. Feature wavelengths were selected through correlation analysis, and six feature parameters including maximum length, maximum width, length-to-width ratio, roundness, area, and perimeter were extracted from the corresponding images to construct a linear discriminant analysis (LDA) model. Compared to image extraction, the LDA model based on image feature parameters (785.6 nm channel) performed the best, achieving 100% precision, 100% recall, and 97.56% accuracy. Sugiyama et al. [14] successfully visualized foreign bodies (leaves and stems) in frozen blueberries using near-infrared (NIR) spectral imaging and discriminant analysis. Ok et al. [15] detected foreign bodies concealed in dry food using a high-resolution raster scan imaging system operating in the sub-terahertz wave range of 210 GHz.

These studies have achieved the identification of foreign bodies using non-destructive detection methods. In this study, we not only utilized hyperspectral technology for foreign body identification but also combined it with target detection techniques to accurately locate the chili pepper foreign bodies. This will provide the necessary information for regulatory authorities to remove the foreign bodies from the products.

## 2. Materials and Methods

### 2.1. Sample Preparation

The chili pepper samples used in the experiments were obtained from the local vegetable wholesale market. Based on commonly encountered foreign bodies in actual production, we selected chili peppers and four types of foreign bodies (red stones, red plastics, red fabrics, and red paper) as shown in Figure 1.

### 2.2. Hyperspectral Image Acquisition

The samples were scanned and acquired using the GaiaSorter Hyperspectral sorter, manufactured by Dualix (Zolix, Beijing, China) in Figure 2. The sorter was equipped with the SpecVIEW software (DualixSpectral Imaging, Wuxi, China). The high-spectral imaging system used a camera with a spectral range of 400 nm–1000 nm, covering both visible light and a portion of near-infrared light. The camera had 176 bands and a spectral resolution of 3.4 nm. The sorter included four halogen lamps, a horizontally movable platform, and a high-precision stepper motor. All these components were enclosed in a dark box measuring 500 mm × 1100 mm × 1800 mm, designed to eliminate the influence of external ambient light.

Prior to capturing high-spectral images of the samples, a preheating period of approximately half an hour was required to avoid the initial instability of the instrument and minimize baseline drift. Through multiple parameter optimizations, the camera exposure time was set to 8.1 ms, and the forward speed of the displacement platform was set to 1 cm/s. This speed ensured that there would be no distortion or deformation in the scanning direction of the camera. Additionally, a platform rollback time of 2.5 cm/s was set to save sample collection time. Once all the parameters were set, the samples were placed on the displacement platform, and the SpecVIEW software (DualixSpectral Imaging, Wuxi, China) was used to coordinate the control of the stepper motor and the high-spectral imaging system for data acquisition.

### 2.3. Correction of Hyperspectral Images

Black–white board calibration is an important method in the calibration of hyperspectral images, which involves the inclusion of black and white reference objects in the images to improve their quantitative accuracy and precision [16]. During the acquisition of hyperspectral images, factors such as the sensor’s nonlinear response, spectral fluctuations, and background noise can introduce errors in the image intensity and spectral response, thereby affecting subsequent data analysis and applications. The black–white board calibration method addresses this issue by incorporating black and white reference objects in the hyperspectral images, transforming them into reflectance or radiance images, thus enhancing the quantitative accuracy and precision of the images. Specifically, the method exploits the distinct spectral responses of the black and white boards, utilizing the comparison between their respective images to correct spectral deviations and noise in the hyperspectral image. During the black–white board calibration process, the calibration software’s interface is used to position the calibration white board directly beneath the end point to capture the white board image. Subsequently, the lens cover is closed, and a black board image is scanned.

The calculation formula of black and white calibration is:(1)$$I_{\mathrm{corr}}=\frac{I-I_{\mathrm{black}}}{I_{\mathrm{white}}-I_{\mathrm{dark}}}$$

In Formula (1), I is the original image data; I_black_ is all black image data; I_white_ is all white image data; and I_corr_ is the corrected image data. After all the original image data were calibrated in black and white, the subsequent analysis could be carried out.

### 2.4. Acquisition of Model Training Samples

For SVM, RF, BP, and other algorithms, 6000 pixels or spectral lines were selected from the high-resolution spectral data for training in Figure 3. First, a mask image was created, and then the coordinates of the selected pixels in the hyperspectral data were obtained using the mask image. Finally, the corresponding spectral data were extracted by performing index calculations in the spectral cube based on the obtained coordinates in Figure 4.

### 2.5. Methods for Establishing Discriminant Models

Supervised classification methods, including random forest, support vector machine, and minimum distance, were adopted for pixel level classification.

Random forest is an ensemble learning algorithm based on decision trees, which combines the results of multiple decision trees to perform classification or regression [17]. The core of the random forest algorithm is the random selection of training data and features to reduce overfitting. When constructing each decision tree, random forest randomly selects a subset of data from the training set and performs random feature selection. This randomness enables random forest to effectively address the overfitting problem and improve the model’s generalization ability. Additionally, random forest can evaluate the contribution of each feature to the classification results, aiding in feature selection for users. Random forest demonstrates high accuracy and robustness in practical applications and is widely used for classification and regression tasks in various fields.

Support vector machine (SVM) is a binary classification model based on statistical learning theory. It can transform linearly inseparable samples into linearly separable forms by mapping them to a high-dimensional space [18]. The goal of SVM is to find an optimal hyperplane that can separate positive and negative samples, while maximizing the distance between the hyperplane and the closest samples. SVM can be categorized into linear SVM and non-linear SVM. Linear SVM seeks the best linear classification hyperplane in the original feature space, while non-linear SVM maps the original data to a high-dimensional space using a kernel function and then finds the optimal linear classification hyperplane in that space. SVM exhibits excellent classification performance and generalization ability, particularly in handling small-sample and high-dimensional data [19].

The minimum distance method is a classification approach based on distance measurements. It compares the distances between the unlabeled samples and the samples from known classes and assigns the unlabeled samples to the class with the closest distance [20]. The core idea of the minimum distance classification method is to treat the distances between the unlabeled samples and the known samples as a measure of dissimilarity, thereby transforming the classification problem into a distance measurement problem. The minimum distance classification method includes different distance measurement methods such as Euclidean distance, Manhattan distance, and Chebyshev distance. Euclidean distance is the most commonly used distance measurement method, which considers the errors in each dimension equally and is suitable for classification problems with continuous features.

### 2.6. Object Detection Algorithm

R-CNN (regions with CNN features), Faster R-CNN (region-based convolutional neural network), and YOLOv5 (you only look once version 5) are object detection methods based on convolutional neural networks (CNN). R-CNN decomposes the object detection problem into region proposal and classification [21], while Faster R-CNN introduces the region proposal network (RPN) for generating region proposals [22]. YOLOv5 is a fast and efficient method that employs an anchor-free approach and adaptive multi-scale training [23]. These methods combine deep features with traditional algorithms, offering advantages such as high accuracy, scalability, fast detection speed, and improved accuracy. They have been widely applied in various domains, including image and video object detection.

### 2.7. Evaluation Metrics for Classification Algorithms

Commonly used evaluation metrics for classification algorithms include accuracy, per-class accuracy, recall, specificity, precision, and F1 score. Accuracy measures the proportion of correctly classified samples, while per-class accuracy measures the proportion of correctly classified samples for each class. Recall measures the ability to identify positive samples, while specificity measures the ability to identify negative samples. Precision measures the accuracy of positive sample predictions, and F1 score is a metric that combines recall and precision. These metrics can be selected and weighted based on specific requirements. For imbalanced data, precision and recall may be more important, while overall accuracy is suitable for balanced datasets. The formulas for these metrics are provided accordingly.
Accuracy = (TP_1_ +TP_2_+…+TPn)/(All)(2)
Per-class Accuracy = (TP +TN)/(TP + FP +TN + FN)(3)
Recall = TP/(TP + FN)(4)
Precision = TP/(TP + FP)(5)
Specificity = TN/(TN + FP)(6)
F1 Score = (2 × P × R)/(P + R)(7)
where TPn is the number of correctly classified samples for each class, All is the total number of test samples, TP is true positive, TN is true negative, FP is false positive, FN is false negative, P is precision, and R is recall.

### 2.8. Evaluation Metrics for Object Detection Algorithms

The evaluation metrics for object detection algorithms include the following aspects:

Accuracy measures the correctness of the model’s detection results, specifically the proportion of correctly predicted positive samples.

Mean Average Precision (mAP) is one of the widely used evaluation metrics in object detection. It evaluates the overall performance of the model by calculating the area under the precision-recall curve for different classes.

Speed is a metric that measures the efficiency of the model’s execution, including the inference time or frame rate. Faster detection speed is an important consideration in real-time applications or large-scale dataset processing.

### 2.9. Data Processing Flow

The methods employed in this study involve the following steps: First, a set of high-resolution hyperspectral images was acquired from the field as samples. These images contained spectral information from different scenes. Next, we applied pattern recognition classification methods for pixel-level classification of the hyperspectral images. By comparing the performance metrics of different classification methods, we selected the random forest classification method as the best model. Random forest exhibits good classification performance and robustness when dealing with high-dimensional data and complex scenes.

For object detection and annotation, we used the LabelImg tool to generate annotated images. This tool allowed us to manually mark the target objects in the images and provided accurate label information for subsequent training and evaluation.

Subsequently, we conducted the training and evaluation of object detection algorithms and identified YoloV5 as the optimal model. YoloV5 is a fast and accurate object detection algorithm capable of effectively identifying and localizing target objects in complex scenes. With the trained YoloV5 model, we were able to recognize mixed foreign objects and obtain their planar coordinate information.

Finally, we transmitted the coordinates of the foreign objects to the execution unit through an industrial communication network to achieve automatic removal of the foreign objects. This step ensured the efficiency and accuracy of foreign object removal.

The entire process involves multiple technical methods and algorithm models, including hyperspectral image acquisition, pixel-level classification, and object detection. By integrating these methods in Figure 5, we successfully achieved the efficient and accurate identification and removal of foreign objects. This research provides valuable technical and methodological references for hyperspectral image processing and applications and holds broad potential for various applications.

## 3. Results and Discussion

### 3.1. Spectral Characteristic Analysis

After establishing the spectral pattern recognition models, we performed pixel-level classification on the hyperspectral images and compared the classification results of the random forest method (Figure 6A), minimum distance method (Figure 6B), and support vector machine (Figure 6C), as shown in Figure 6. From the figure, it can be observed that the random forest method effectively differentiated pixels of each class, resulting in an overall better performance. The minimum distance method exhibited class overlap, leading to suboptimal classification results. The performance of the support vector machine method fell in between, demonstrating a moderate performance.

Next, we conducted a more rigorous quantitative analysis to further evaluate the classification performance. This involved the use of evaluation metrics such as accuracy, recall, and F1 score to objectively quantify the models’ performances. Through quantitative analysis, we could more accurately assess the performance of each classification method and provide a scientific basis for selecting the optimal classification model.

By performing pixel-level classification on the hyperspectral images and conducting subsequent scientific quantitative analysis, we gained a better understanding of the effectiveness of various classification methods. This provided a reliable foundation and guidance for subsequent hyperspectral image processing and analysis.

### 3.2. Evaluation of Different Spectral Classification Algorithms

A quantitative test method was used to test the classification model. First, 5000 pixels were randomly selected from the test sample as the test set, including 1000 pixels for each category, and the performance of the three classification models on the test set was calculated. The overall accuracy of random forest was 86% higher than support vector machine and the minimum distance model in Figure 7. In the confusion matrix shown in Figure 8, it is clear to see the number of correct classifications for each category and the misclassifications where a category is classified into one of the other five categories. Meanwhile, by analyzing other indicators of classification effect, as shown in Figure 9, it could be seen that random also had a good classification index on a single category.

### 3.3. Labeling of the Detected Object

When annotating the enhanced samples, the LabelImg was used in Figure 10. This tool enables manual annotation of the target objects in the samples. Prior to annotation, five predefined categories were set, namely chili, paper, stone, plastic, and fabric.

Using the LabelImg tool, the sample image was opened, and the appropriate class label was selected in the tool’s interface. The position and boundaries of the target were marked by drawing bounding boxes around the objects. Each bounding box represented one target object and was associated with a predefined class. Careful observation of the target objects in the image was required during the annotation process to ensure accuracy and consistency. The tool provides features such as zooming and panning to aid in better recognition and annotation of the targets. Once the annotation was completed, the LabelImg tool generated annotation files associated with each target object, containing information about the target class and the location of the bounding box. These annotation files were used for training and evaluating the object detection algorithm and played a crucial role in subsequent tasks such as target recognition and classification.

By annotating the enhanced samples using the LabelImg tool and labeling the target objects according to the predefined five categories, accurate training data were provided for subsequent object detection and classification tasks, thereby aiding in improving the performance and accuracy of the algorithm.

### 3.4. Detection of Foreign Objects Using Object Detection Algorithms

After the spectral information of the hyperspectrum was used to strengthen the target classifier, the easily confused objects became easy to distinguish. Under this premise, small samples were used to train the model—300 labeled samples of each type, and 1500 labeled samples in total.

By comparing the experimental results, as shown in Table 1, we found that YOLOv5 outperformed Faster R-CNN and R-CNN in terms of performance metrics. YOLOv5 achieved higher accuracy and recall in the object detection task, while having a lower false positive rate. Figure 11 shows the detection results in the mixed state, while Figure 12, Figure 13, Figure 14 and Figure 15 demonstrate the detection results in the overlapping state. In addition, YOLOv5 also exhibited faster execution speed and could complete detection operations at the millisecond level, which is very valuable for real-time applications.

### 3.5. Calculating the Geometric Coordinates of foreign Objects Using the Results of Object Detection

By using the YoloV5 object detection algorithm, we were not only able to accurately identify foreign objects but also obtain the centroid coordinates of these objects. The purpose of this foreign object detection was to facilitate batch removal of these objects. In the production line workflow, the scanned samples first passed through a hyperspectral detection device, where the computer captured the corresponding detection images. Subsequently, by applying the YoloV5 algorithm to analyze these images, we could obtain the spatial coordinates of the foreign objects, as shown in Figure 16.

Once we had obtained the coordinates of the foreign objects, we could transmit this positional information to subsequent execution mechanisms, such as XY-type or parallel-type robots, through an industrial network. These robots could then perform the necessary operations to remove the foreign objects based on the provided coordinates. This enabled automated removal of foreign objects in the production line.

### 3.6. Statistical Results of Measurement Errors of Object Detection Methods

For 100 sets of samples, we performed coordinate measurements using a digital vernier caliper and compared the results with those obtained from the object detection algorithm. By comparing the measurements, we found that the maximum error in the X coordinate was 0.61 mm, while the maximum error in the Y coordinate was 0.58 mm. To better evaluate the error situation, we calculated the average error values after taking the absolute values. The results showed an average error of 0.33 mm for the X coordinate and 0.3 mm for the Y coordinate. By plotting these results, we could visually observe the distribution of measurement errors in Figure 17.

## 4. Conclusions

In this study, we investigated the application of hyperspectral image processing and object detection methods in foreign object recognition and removal. By acquiring hyperspectral images of the samples and performing pixel-level classification using pattern recognition techniques, we identified random forest classification as the optimal model and generated annotated images using the LabelImg tool. Subsequently, we trained and evaluated object detection algorithms, ultimately selecting YoloV5 as the best model. By applying the YoloV5 object detection algorithm, we successfully achieved a recognition accuracy of 96% for mixed foreign objects and obtained the centroid coordinates of the objects with a positioning accuracy of ±0.61 mm. Through an industrial communication network, we were able to transmit these coordinates to execution mechanisms for automated foreign object removal. This comprehensive approach combines different technical methods and algorithm models in various stages, including hyperspectral image processing, pixel-level classification, and object detection, providing a viable solution for efficient and accurate foreign object recognition and removal. Future research can focus on further optimizing algorithm models and industrial execution mechanisms to improve the efficiency and reliability of foreign object detection and removal, offering better solutions for industrial production.

## Figures and Tables

**Figure 1 foods-12-02618-f001:**
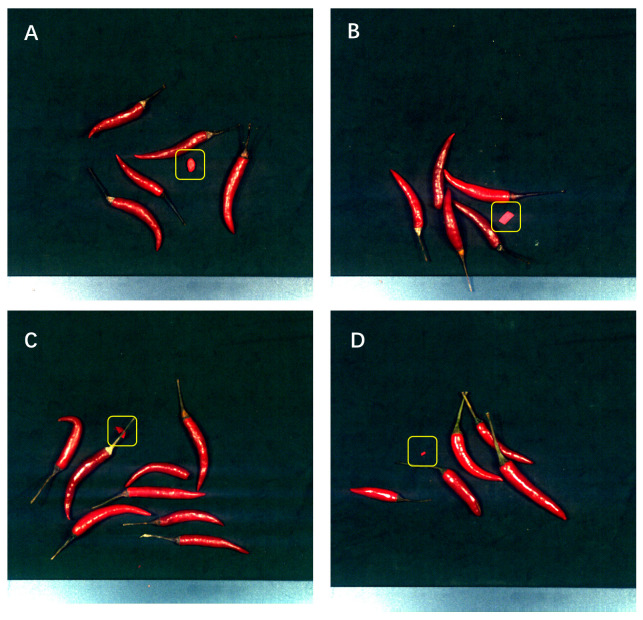
(**A**). mixed stone, (**B**). mixed paper, (**C**). mixed fabric, (**D**). mixed plastic.

**Figure 2 foods-12-02618-f002:**
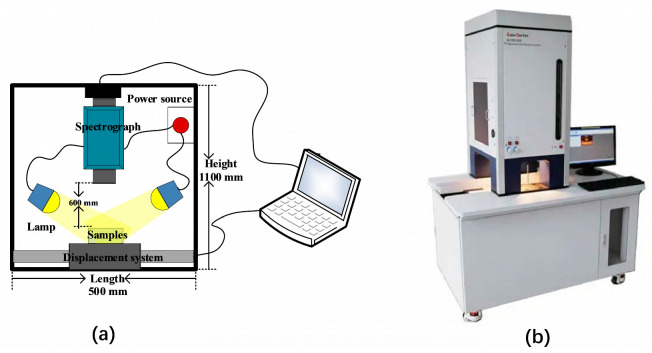
(**a**) Sketch map of hyperspectral system. (**b**) Real photos of the Hypersectral Camera.

**Figure 3 foods-12-02618-f003:**
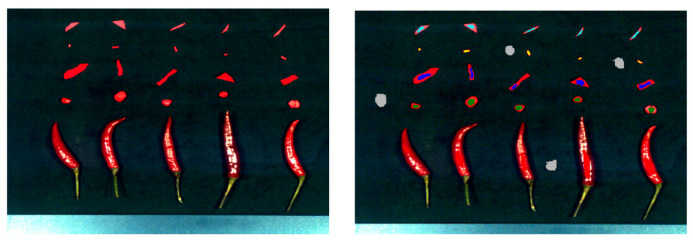
Select the region of interest and extract the spectrum.

**Figure 4 foods-12-02618-f004:**
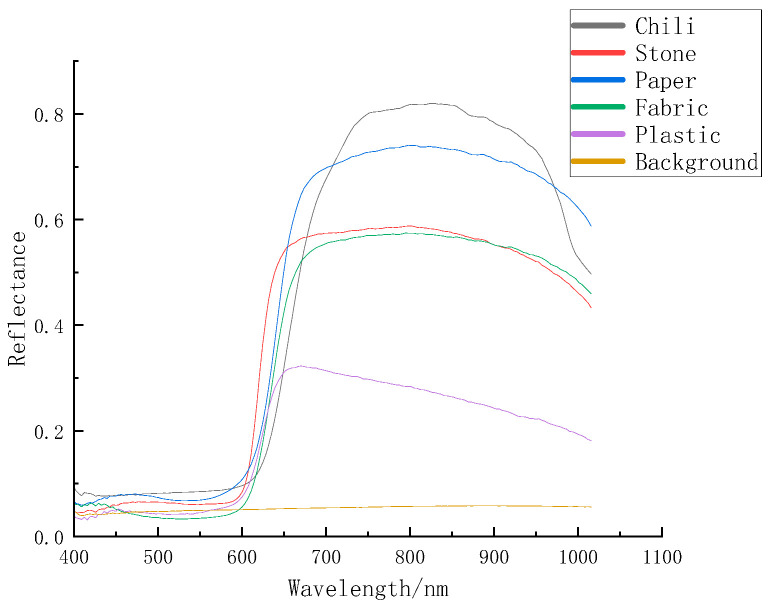
Comparison of spectral curves of six substances.

**Figure 5 foods-12-02618-f005:**
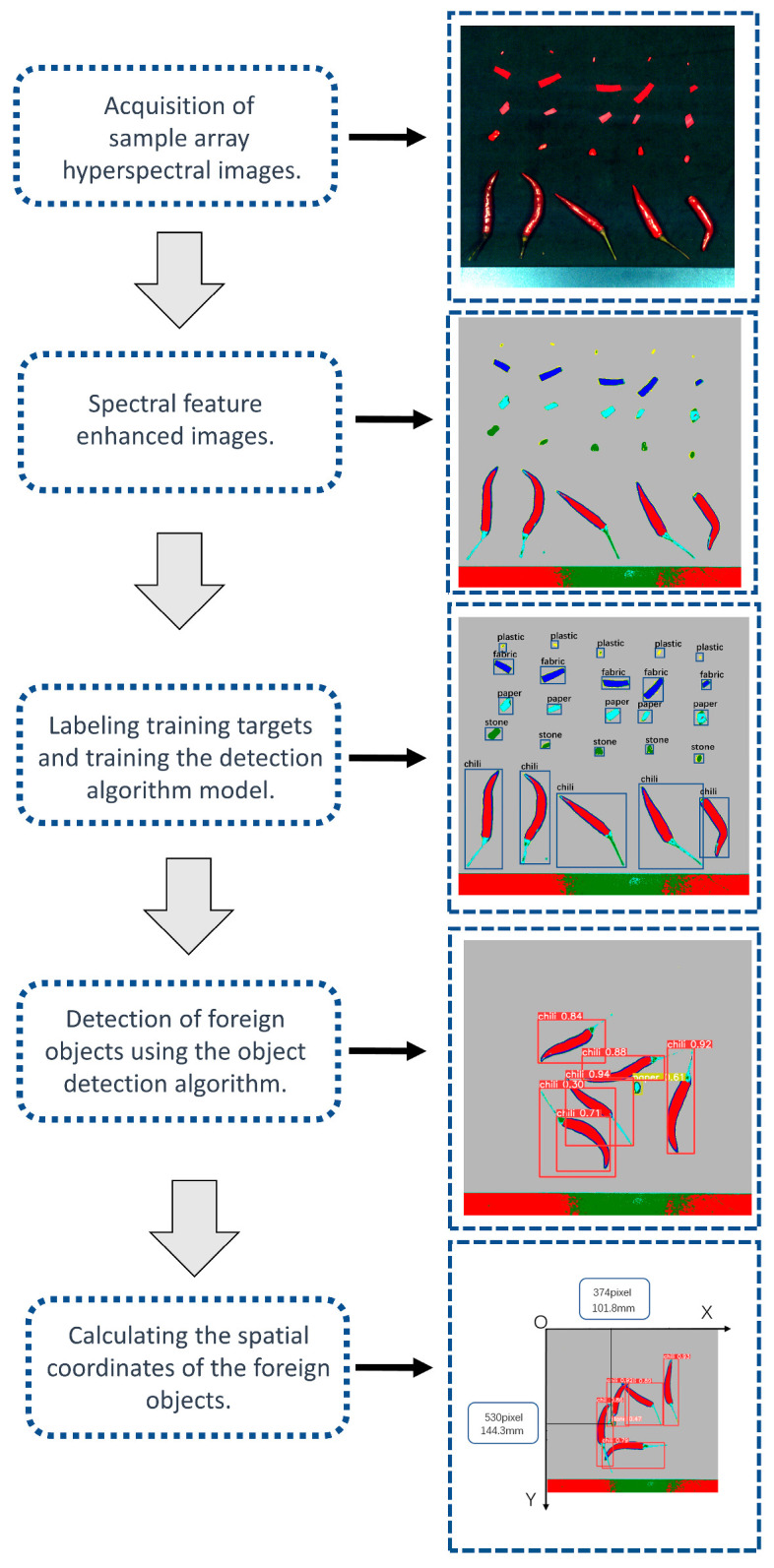
Data processing flow.

**Figure 6 foods-12-02618-f006:**
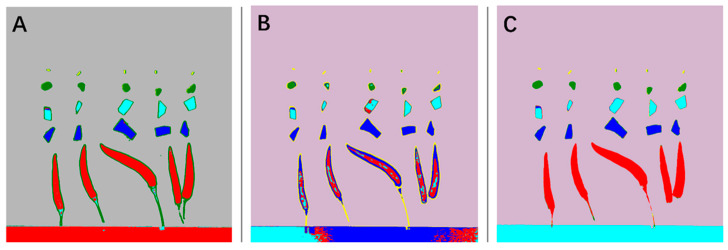
Visualization of pixel-level classification. (**A**). Random forest, (**B**). minimum distance, (**C**). SVM.

**Figure 7 foods-12-02618-f007:**
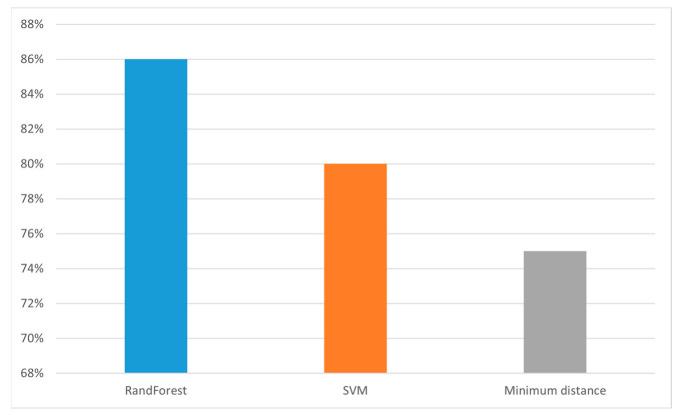
Comparison of overall accuracy rates of classification algorithms.

**Figure 8 foods-12-02618-f008:**
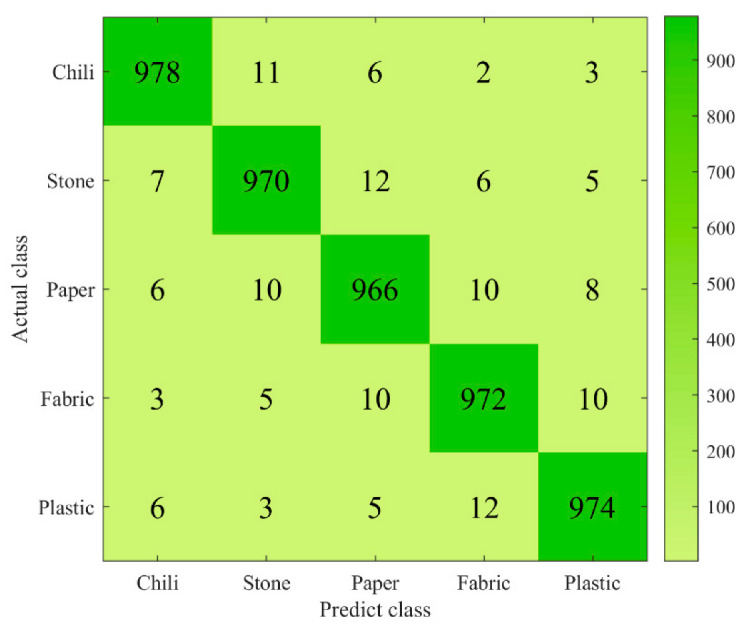
Confusion matrix of random forest classification model.

**Figure 9 foods-12-02618-f009:**
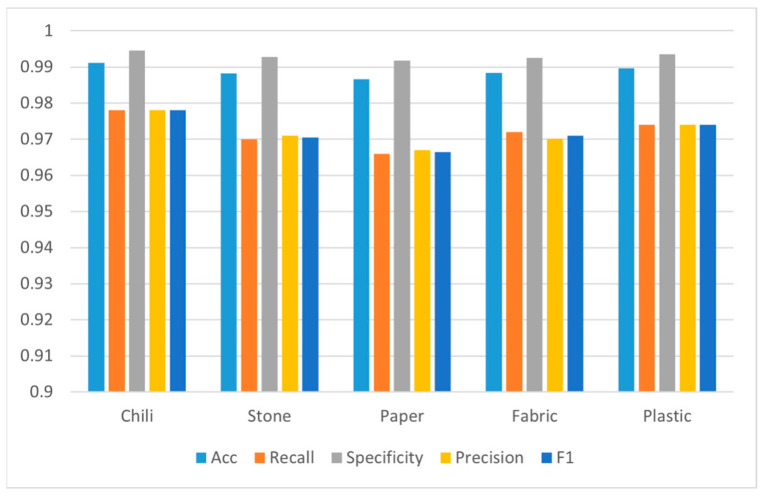
Evaluation metrics of random forest classification model on different categories.

**Figure 10 foods-12-02618-f010:**
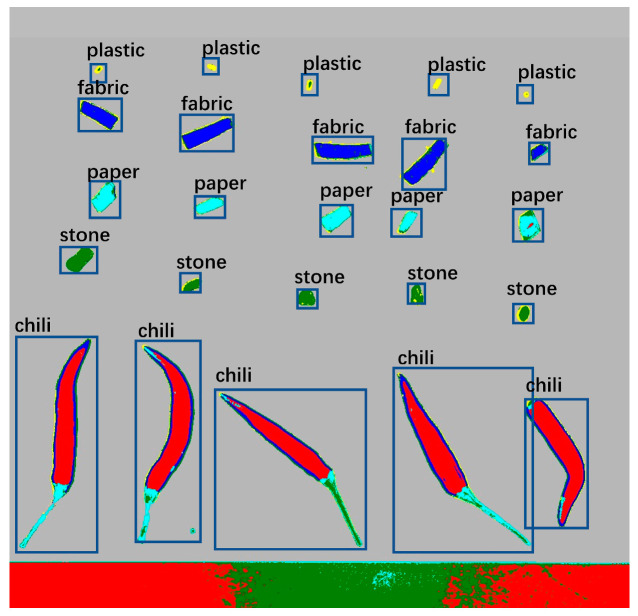
Five classes of samples are labeled.

**Figure 11 foods-12-02618-f011:**
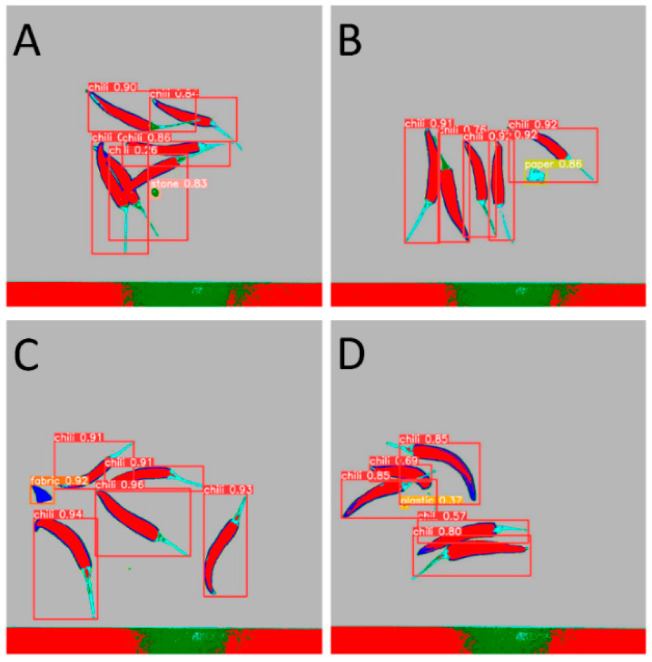
Detection of mixed samples by Yolo V5. (**A**). Stone, (**B**). Paper, (**C**). Fabric, (**D**). Plastic.

**Figure 12 foods-12-02618-f012:**
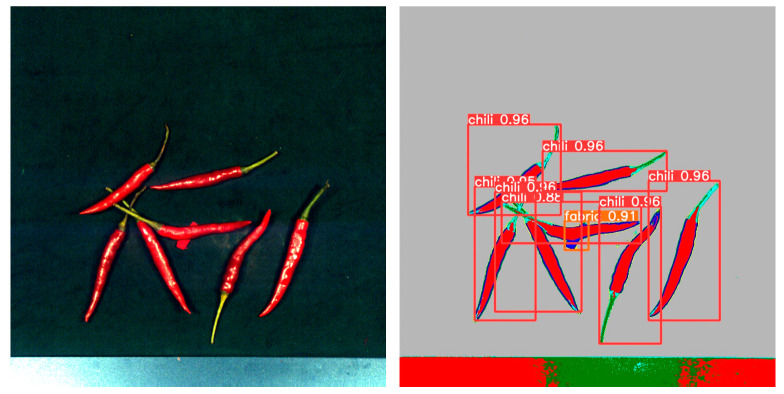
The case of fabric overlap.

**Figure 13 foods-12-02618-f013:**
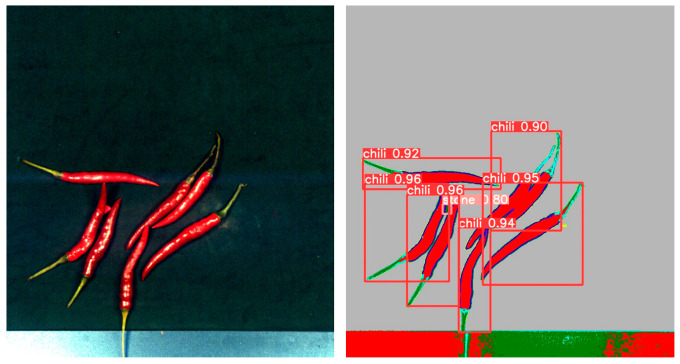
The case of stone overlap.

**Figure 14 foods-12-02618-f014:**
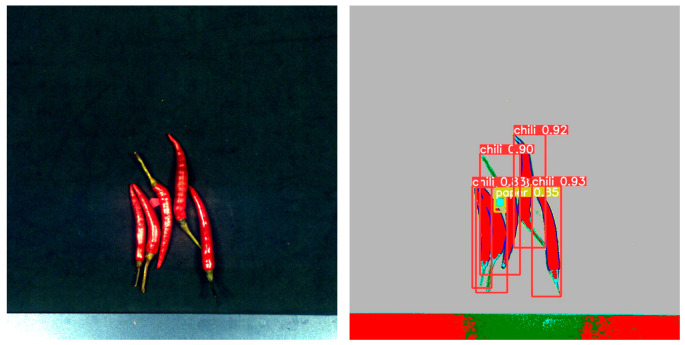
The case of paper overlap.

**Figure 15 foods-12-02618-f015:**
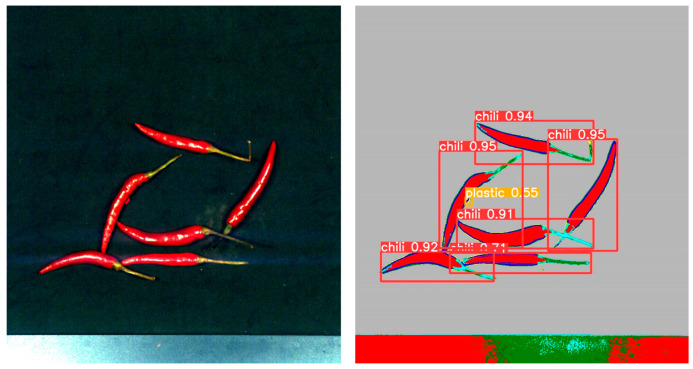
The case of plastic overlap.

**Figure 16 foods-12-02618-f016:**
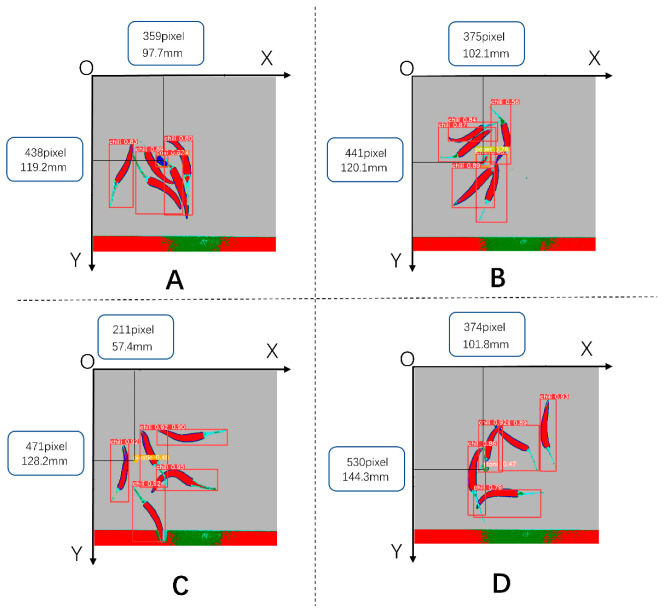
Calculation of the spatial coordinates of the target foreign body. (**A**). Fabric, (**B**). Paper, (**C**). Plastic, (**D**). Stone.

**Figure 17 foods-12-02618-f017:**
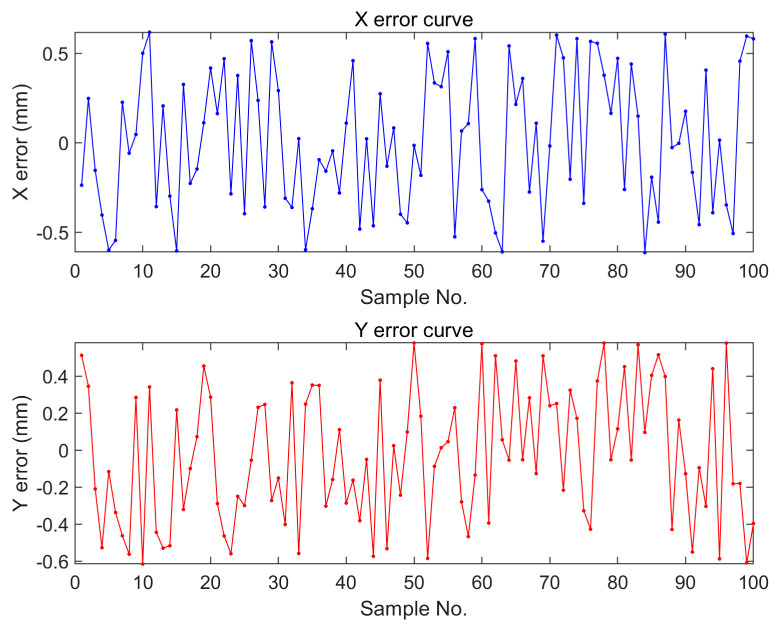
X-coordinate error and y-coordinate error distribution of 100 samples.

**Table 1 foods-12-02618-t001:** Evaluation of object detection algorithms.

Object Detection Algorithm	mAp	Accuracy	Average Time
Faster-CNN	0.75	85%	50 ms
R-CNN	0.65	75%	1.2 s
Yolo V5	0.93	96%	12 ms

## Data Availability

The datasets generated for this study are available from the authors.

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
