# Peer review of "Detection of Chili Foreign Objects Using Hyperspectral Imaging Combined with Chemometric and Target Detection Algorithms"

_foods, 2023, doi:10.3390/foods12132618_

Round 1

Reviewer 1 Report

1. Most of the objects(pepper, foreign body) tested in manuscript are in a scattered state without overlapping.

   In this case, it is considered that coventional RGB cameras (including hyperspectral cameras) also easily detect. 

   Therefore, authors shall show performance results with objects overlapping or partially exposed.

2. Did the authors adopt YOLO_v5 as a detection algorithm without modification?

   If so, it is considered that the contribution is low because a widely known technology is used.

3. Also, YOLO_v5 was announced in 2020, so it is difficult to see it as the state-of-the-art technology.

   It is necessary to present the ideas of the authors only.

4. In line 388, the authors address that the positioning accuracy of foreign objects was 0.5 mm, but there is no actual test content.

   The authors shall present accurate measurements and average results.

5. In section 2.2, it is recommended to add detailed specifications and real photos of the Hypersectral Camera.

6. In Line 266, it is recommended to explain which labeling tool was used.

7. The resolution of figure 5, 10, 11, and 12 is too low. Especailly, the labels in the figures are not visible.

Reviewer 2 Report

The following improvements are suggested:

title: the title is non-informative. Suggest to add imaging after hyperspectral, and "chemometric" before target detection

Abtract: Foods is not using subheadings in the abstract, delete

Figure 1: the defects are not visible, improve

Section 2.2: add more details on measurement principle and device (manufacurer, software version etc). Hyperspectral means visible and part of infrared? Define and specify.

Sections 2.5-2.8: This is mostly textbook knowledge and should be much condensed. Citations to methods would be sufficient.

Figure 7: add unit, %?

References: check for style according to template, e.g. journal abbreviations, DOI missing

Round 2

Reviewer 1 Report

The manuscript is much improved by reflecting reviewer's comments.